# Identity Hides in Darkness: Learning Feature Discovery Transformer for Nighttime Person Re-Identification

**DOI:** 10.3390/s25030862

**Published:** 2025-01-31

**Authors:** Xin Yuan, Ying He, Guozhu Hao

**Affiliations:** 1School of Computer Science and Technology, Wuhan University of Science and Technology, Wuhan 430065, China; xinyuan@wust.edu.cn (X.Y.); hy@wust.edu.cn (Y.H.); 2Hubei Province Key Laboratory of Intelligent Information Processing and Real-Time Industrial System, Wuhan University of Science and Technology, Wuhan 430065, China; 3Hubei Key Laboratory of Inland Shipping Technology, Wuhan 430063, China; 4School of Navigation, Wuhan University of Technology, Wuhan 430063, China

**Keywords:** low illumination, person re-identification, low-frequency information, auxiliary learning

## Abstract

Person re-identification (Re-ID) aims to retrieve all images of the specific person captured by non-overlapping cameras and scenarios. Regardless of the significant success achieved by daytime person Re-ID methods, they will perform poorly due to the degraded imaging quality under low-light conditions. Therefore, some works attempt to synthesize low-light images to explore the challenges in the nighttime, which omits the fact that synthetic images may not realistically reflect the challenges of person Re-ID at night. Moreover, other works follow the “enhancement-then-match” manner, but it is still hard to capture discriminative identity features owing to learning enlarged irrelevant noise for identifying pedestrians. To this end, we propose a novel nighttime person Re-ID method, termed Feature Discovery Transformer (FDT), explicitly capturing the pedestrian identity information hidden in darkness at night. More specifically, the proposed FDT model contains two novel modules: the Frequency-wise Reconstruction Module (FRM) and the Attribute Guide Module (AGM). In particular, to reduce noise disturbance and discover pedestrian identity details, the FRM utilizes the Discrete Haar Wavelet Transform to acquire the high- and low-frequency components for learning person features. Furthermore, to avoid high-frequency components being over-smoothed by low-frequency ones, we propose a novel Normalized Contrastive Loss (NCL) to help the model obtain the identity details in high-frequency components for extracting discriminative person features. Then, to further decrease the negative bias caused by appearance-irrelevant features and enhance the pedestrian identity features, the AGM improves the robustness of the learned features by integrating the auxiliary information, i.e., camera ID and viewpoint. Extensive experimental results demonstrate that our proposed FDT model can achieve state-of-the-art performance on two realistic nighttime person Re-ID benchmarks, i.e., Night600 and ${\mathrm{RGBNT}201}_{rgb}$ datasets.

## 1. Introduction

Person re-identification (Re-ID) is a hot computer vision task of matching a person of interest across different camera views [1,2]. With urban development and rising demands for public security and forensics, Re-ID research has become a key focus due to the crucial role that pedestrians play in areas like public safety and urban management [3,4]. Earlier person Re-ID methods addressed challenges like viewpoint change [5], clothes-changing [6,7], occlusions [8,9], pose variation [10], etc. With the advances in deep learning techniques and the proposal of large-scale datasets, the accuracy of person Re-ID methods has been progressively improved [11,12]. Notably, almost most of the existing person Re-ID methods focus on daytime scenarios, and most publicly available person Re-ID datasets were collected within a fixed period of daytime [13]. However, these daytime person Re-ID methods cannot be directly applied to nighttime scenarios, as low-light conditions at night deteriorate the image quality, leading to degradation in the person Re-ID performance.

In the early years, there were some attempts to study person Re-ID under low illumination. Since there are no available low-light datasets, a straightforward way is to use data-enhancement techniques (e.g., color jitter and gamma correction) to synthesize low-light images for simulating low-light scenarios [14,15,16,17,18]. To simulate low-light conditions, both Huang et al. [14] and Zeng et al. [16] utilized gamma correction to build new low-light image datasets based on Market-1501 [19] and DukeMTMC-reID [20] as illustrated in Figure 1a, and they experimentally verified that low illumination admittedly degrades the person Re-ID performance [15]. To save costs for labeling on different illumination scales, Zhang et al. [17] attempted to transform the test images with different illumination scales to the same uniform illumination scale as the training images. Unlike these methods, Huang et al. [18] designed a new degradation invariance learning framework, focusing on low-quality images, not just only low-light images. Additionally, Bak et al. [21] and Xiang et al. [22] studied the impact of varying illumination on the Re-ID system performance using synthesized person Re-ID datasets, which were generated with different illumination scales through a 3D game engine [23], as shown in Figure 1b. Although the above person Re-ID methods have investigated the issue of low illumination, they are all based on synthetic data, which makes it hard to simulate and reflect the challenges of realistic nighttime scenarios.

To investigate the challenges of realistic nighttime person Re-ID with low illumination, recent works have constructed several person Re-ID datasets collected from realistic nighttime scenarios [24,25,26,27,28]. Specifically, Zhang et al. [26] first focused on night scenarios and designed a new nighttime person Re-ID dataset called KnightReid. The KnightReid contains three camera views, but most of its pedestrian identities are derived from only two cameras. Penate-Sanchez et al. [27] used a sporting event as the scenario and collected the TGC20ReId dataset in the running sports domain, which contains both daytime and nighttime scenarios, but the clarity and brightness of the images in nighttime scenarios are generally higher due to the presence of brighter lights on the runway [29]. To facilitate the research of nighttime person Re-ID and better reflect the challenges of real nighttime scenarios, Lu et al. [24] contributed a realistic nighttime person Re-ID dataset named Night600, consisting of a rich set of low-illumination conditions with different viewpoints. In addition, some datasets (e.g., LLCM [28] and RGBNT201 [25] presented in Figure 1c) have been proposed to primarily focus on the nighttime cross-modality problem [24,30]. In this paper, we mainly concentrate on the RGB modality challenges in nighttime scenarios.

In order to investigate nighttime person Re-ID with only RGB modality data, existing methods adopt a most straightforward solution, i.e., the “enhancement-then-match” manner. As shown in Figure 2a, the “enhancement-then-match” methods can be classified into two categories: (I) one branch and (II) dual branch. For the one-branch method, it involves two stages: (1) Firstly, improving the visibility of the images through Enhancement Net in stage I [31]; (2) Then, feeding the relighted images to DNN (i.e., Re-ID model) in stage II [32]. For the dual-branch methods, it is an end-to-end way, which integrates relighting and Re-ID into a single model [24]. Unfortunately, both types of methods require two models, i.e., Enhancement Net and Re-ID networks, which significantly increases the complexity and time costs in real-world applications.

Different from these methods, this paper considers a simple but effective “enhancement drop” solution. As illustrated in Figure 2b, we propose the Feature Discovery Transformer (FDT) for nighttime person Re-ID to explicitly capture the pedestrian identity information hidden in darkness at night. Concretely, the FDT model mainly has the Frequency-wise Reconstruction Module (FRM) and the Attribute Guide Module (AGM). The FRM is proposed to separate the high- and low-frequency components by utilizing the Discrete Wavelet Transform (DWT). The low-frequency component captures the main structure of the image, while the high-frequency component contains detailed features. To address poor nighttime image quality affecting pedestrian identity information, transformer-based models [32,33,34] that focus on global features are employed. Additionally, the FRM quantizes the low-frequency component to prevent overfitting. The high-frequency component holds detailed yet noisy information that often gets lost during learning. Additionally, nighttime images often contain similar irrelevant information across images of the same category. To prevent these challenges, a new Normalized Contrastive Loss (NCL) is introduced to avoid its over-smoothing by low-frequency components. By emphasizing the high-frequency component, it reduces the distance between samples of the same category in the embedding space while increasing the distance between different categories, enhancing pedestrian identity discrimination. Moreover, to reduce the negative influence of appearance-irrelevant features and enhance the pedestrian identity features, we use the AGM to integrate the existing attribute information (i.e., camera ID and viewpoint) for improving the robustness of the learned pedestrian features.

Furthermore, the interactions among the aforementioned three components are described as follows. In the auxiliary branch, FRM is mainly responsible for processing the features extracted from low-frequency components. Meanwhile, NCL is employed to prevent these features from being over-smoothed by high-frequency components. In the main branch, AGM serves to reduce the impact of irrelevant features during the feature extraction process. During the model learning process, the two branches interact with each other through parameter sharing, thereby enhancing the performance. Experimental results have demonstrated that the proposed FDT model achieves promising performance improvement on the nighttime datasets, including Night600 [24] and ${\mathrm{RGBNT}201}_{rgb}$ [25], respectively.

In summary, the contributions of this work are listed as follows:A new Feature Discovery Transformer (FDT) is proposed for the nighttime person Re-ID task, which can better learn the identity features of pedestrians hidden in the dark. To the best of our knowledge, this is the first attempt via the “enhancement drop” solution for nighttime person Re-ID.The Frequency-wise Reconstruction Module (FRM) processes image frequencies, quantizing low frequencies to enhance global pedestrian features. Simultaneously, the Normalized Contrastive Loss (NCL) prevents high-frequency over-smoothing, capturing detailed high-frequency information to distinguish pedestrian identities.The Attribute Guide Module (AGM) is introduced to integrate auxiliary information, thereby enhancing the robustness of the extracted pedestrian features.Experimental results on the Night600 [24] and ${\mathrm{RGBNT}201}_{rgb}$ [25] datasets demonstrate the effectiveness of our FDT model, achieving state-of-the-art performance in the nighttime person Re-ID task.

## 2. Related Works

### 2.1. Person Re-Identification

Current person Re-ID methods mainly concentrate on enhancing feature learning and metric learning, thereby attaining notable performance on publicly available person Re-ID datasets [11,35]. In particular, within the domain of feature learning, the majority of person Re-ID methods involve the adaptation of CNN-based and Transformer-based models to address the challenges inherent in person Re-ID tasks.

For CNN-based methods, Luo et al. [36] introduced a strong CNN-based baseline for extracting discriminative features. Ye et al. [37] developed the AGW baseline, which uses a Non-local Attention Block to enhance model performance through attention mechanisms. Recent methods have shifted towards optimizing Re-ID models to better handle challenges like viewpoint and clothing changes, occlusions, and pose variations. Zhang et al. [38] enhanced Re-ID performance for pedestrians in varied outfits by integrating biometric features and domain adaptation. Zhou et al. [39] combined local and global features, but CNN methods struggle to differentiate pedestrian features in nighttime scenarios due to their focus on small areas.

With the development of transformers, visual transformers (ViTs) have been effectively applied to person Re-ID tasks. He et al. [32] introduced a ViT-based Re-ID framework that outperforms leading CNN-based methods. Additionally, notable transformer architectures like AAformer [33] and LA-Transformer [34] have been created for supervised Re-ID tasks. AAformer uses part labels to help the transformer learn part-specific features, while LA-Transformer uses a PCB-like approach for effective part-level feature extraction. PASS [34] employs ViTs to capture fine-grained details essential for Re-ID. In addition, some methods still integrate transformer layers into CNN backbones to combine hierarchical features and align local features. However, these approaches still perform poorly at night, highlighting the need for further improvements.

### 2.2. Nighttime Person Re-Identification

Nighttime person Re-ID aims to recognize individuals across different cameras at night, facing challenges like low-light, background clutter, varying camera angles, and resolution problems in surveillance. Research has increasingly focused on tackling illumination and low-light issues. Kviatkovsky et al. [40] used a color distribution structure to create color-invariant features, addressing differences in illumination. Conversely, Yu et al. [41] argued that color features are unreliable with illumination changes and focused on learning fabric color features based on localized skeleton information. Bhuiyan et al. [42] tackled device lighting inconsistencies by developing a robust light transfer function to standardize illumination. Building on this, Zeng et al. [16] extended the study to nine light scales, achieving a unified lighting condition with a single scale. After that, Zhang et al. [17] expanded the light scale range and established a consistent, uniform light scale. To address the low-illumination issue, some methods attempted to utilize gamma-correction techniques [15,16,17,18] or nighttime person datasets synthesized by 3D game engines [21,22]. The data images obtained through gamma correction possess a uniform overall illumination, rendering it arduous to model the challenges associated with localized over-darkness or overexposure. The low-illumination datasets synthesized by 3D game engines feature lighting that emulates a virtual setup, which exhibits a slight disparity from the real-world scenario. Although these methods have significantly progressed in tackling illumination variations in Re-ID, effective solutions for low-light nighttime scenarios are still lacking. This underscores the necessity for ongoing research and innovative strategies to address the challenges of nighttime person Re-ID.

To address Re-ID model performance issues in night scenarios, Zeng et al. [16] used metric learning for low-light conditions, while Huang et al. [14] integrated image recovery and enhancement algorithms with the Re-ID backbone. Zhang et al. [17] developed a model using the difference between light-adjusted and original images for light classification. Oliverio et al. [29] enhanced and de-blurred pedestrian images in low-light. Lu et al. [24] trained a Re-ID model with nighttime images by combining features from light-enhanced and original images, using light distillation to reduce noise. Nevertheless, the method yielded sub-optimal person Re-ID results due to poor low-light image quality and insufficient enhancement for nighttime person Re-ID needs.

### 2.3. Auxiliary Information Learning

To tackle low-light and noise in nighttime data, various auxiliary information learning approaches are used to process and extract pedestrian information for the person Re-ID task. This involves enhancing features specific to individuals, such as attribute and skeleton information, as well as extracting information beyond individual characteristics. The attribute information, unaffected by viewpoint and lighting, is a reliable source for learning person representations. Schumann et al. [43] showed that its automatic detection aids in learning these representations. Lin et al. [13] manually labeled attributes for the Market1501 [19] and DukeMTMC [20] datasets. Li et al. [44] enhanced the image encoder by adding learnable attributes to the text encoder. Zhai et al. [45] improved accuracy by merging attribute data from ChatGPT and datasets. Lee et al. [46] studied long-term clothing changes to derive clothing and accessory attributes. Yet, these methods mainly depend on clear, well-lit images and often need extensive manual annotation to extract attributes. This becomes challenging at night, where factors like viewpoint and camera type limit attribute information, hindering person feature extraction. For information beyond individual characteristics, style transformation techniques are primarily used. Zhou et al. [47] applied style transformation using feature statistics to capture image styles. However, due to poor lighting in nighttime images and similar styles across cameras in the Night600 dataset [24], performance gains were limited. This highlights the need for new strategies in nighttime person Re-ID scenarios.

### 2.4. Image Processing Technologies for Nighttime Re-ID

Nighttime images suffer from low-light conditions and poor quality, making person Re-ID challenging. To improve performance, current methods use image processing techniques to better capture person identities, with a strong emphasis on low-light enhancement. For instance, Guo et al. [31] proposed a curve estimation method for illumination enhancement that does not require paired data, enabling effective enhancement without dataset constraints. Similarly, Liu et al. [48] used Retinex theory to enhance feature visibility in low-light by improving light estimation and denoising. Yang et al. [49] introduced a Transformer-GAN network to address color degradation at night, a common oversight in current methods. These approaches seek to reduce color degradation’s impact on image quality, improving the person Re-ID performance. Advancements in low-light enhancement techniques are vital for better visibility and feature quality in nighttime images, thereby facilitating more accurate person Re-ID results.

However, existing realistic nighttime person Re-ID datasets do not have pair-wise training data for low-light enhancement techniques. Moreover, although unsupervised methods can be utilized, it is difficult to guarantee the visual quality of the enhanced images. Although Asperti et al. [4] performed image generation by tackling the perspective of image noise, it was still limited by the data. Frequency-based methods are advantageous for low-quality images as they do not need paired data. Low-frequency components preserve the main structure, while high-frequency components capture details. Guo et al. [50] introduced a technique using deep CNNs to merge these components, enhancing detail recovery. Yang et al. [51] used the low-frequency spectrum to reduce domain distribution differences and improve image representation consistency. Yao et al. [52] developed a reversible downsampling method to preserve high-frequency details. Furthermore, Ma et al. [53] proposed an innovative combination of frequency-domain information and optical flow to effectively integrate the texture information of high frequencies with the overall image representation. Inspired by the above efforts, we propose a dual approach for nighttime person Re-ID that integrates high-frequency and low-frequency information. This method enhances the extraction of key structural elements and preserves crucial high-frequency details, improving the robustness and accuracy of nighttime person Re-ID.

## 3. Methodology

In this section, we propose a Feature Discovery Transformer (FDT) framework, as illustrated in Figure 3. This framework utilizes a dual-branch architecture, incorporating the Vision Transformer (ViT) as its foundational backbone. Within this configuration, the original main branch is enhanced by the integration of an Attribute Guide Module (AGM) following the introduction of the nighttime pedestrian identity feature. Concurrently, in the auxiliary branch, augmented features are extracted via the Frequency-wise Reconstruction Module (FRM). To ensure that the original main branch does not undermine the efficacy of the FRM during network propagation, the Normalized Contrastive Loss (NCL) is employed.

### 3.1. Transformer-Based Baseline

Compared to those taken under normal lighting conditions, the pedestrian images captured at night typically exhibit lower quality and contain more pedestrian identity information. To enhance the performance of person Re-ID systems, it is crucial to improve image quality to facilitate the extraction of superior features while simultaneously preserving detailed information within the images. Consequently, a Transformer-based Re-ID method, which retains more detailed information than the Convolutional Neural Network (CNN), is chosen to conduct experiments for this work.

Our FDT framework utilizes the Transformer architecture from TransReID-SSL [54] as its backbone. This framework substitutes the standard convolutional stem [55] with an instance batch normalized (IBN) convolutional stem, which is applied atop the ViT following the division of the input image into *N* non-overlapping patches by the patch embedding module. Furthermore, it incorporates IBN to process the input image, drawing inspiration from the CNN-based Re-ID approach, IBN-Net [56]. The feature maps are designed to learn features with illumination invariance during nighttime conditions. This is achieved by employing a patched stem, implemented through a stride-pp×pconvolution (where *p* defaults to 16) on the input image. This approach ensures the performance of the ViT, where *H* and *W* represent the height and width of the image, respectively, and *C* denotes the number of channels. The kernel size of the convolutional stems, which consist of multiple stacked convolutional layers, Batch Normalization (BN), and ReLU activations, is specifically determined through a strategic application of BN. After the initial two shallow convolutional layers, BN is applied to half of the channels, while the other half undergoes a similar treatment subsequently. Additionally, only the BN layer is utilized following the deeper convolutional layers. This approach is informed by the positive outcomes observed from integrating Instance Normalization (IN) and BN, which enhances the invariant representations within the Re-ID learning domain. It is important to note that the channel configuration and step size are maintained consistent with previous works. Furthermore, the final layer incorporates a patched stem, which significantly contributes to optimizing stability and enhancing overall performance. The input image is divided into *n* equally sized patches, denoted as $\left\{x_{i}\in R^{D}|i=1,2,\cdots ,N\right\}$. Each term in $I_{\mathrm{input}}$ is processed through the convolutional stems, followed by a flattening and linear mapping, denoted as $f(x_{i})$. A class label, $x_{\mathrm{cls}}$, is prepended to $\left\{f\left(x_{1}\right);\cdots ;f\left(x_{n}\right)\right\}$ to facilitate the capture of global features. Subsequently, a positional embedding $P\in R^{(N+1)\times D}$ is applied to each patch as follows:(1)$$\begin{array}{cc} I_{input} & =\left[x_{1},x_{2},\cdots ,x_{n}\right] \end{array}$$(2)$$\begin{array}{cc} y & =\left[x_{cls};f\left(x_{1}\right);\cdots ;f\left(x_{n}\right)\right]+P \end{array}$$

The auxiliary information is exclusively integrated at the original image level within the variable *y*. Following this integration, both $y^{0}$ and *y* undergo processing through transformer layers. These layers consist of Multi-head Self-Attention (MSA) and Multi-Layer Perceptron (MLP) modules. The layers are indexed by l∈{1,…,L} and are defined as follows:(3)$$\begin{array}{cc} y^{0} & =y+\lambda_{a}a \end{array}$$(4)$$\begin{array}{cc} y_{att}^{l-1} & =y^{l-1}+MSA\left(LN\left(y^{l-1}\right)\right) \end{array}$$(5)$$\begin{array}{cc} y^{l} & =y_{att}^{l-1}+MLP\left(LN\left(y^{l-1}\right)\right) \end{array}$$ Here, $a\in R^{(N+1)\times D}$ represents the attribute information embedding of the patch. The parameter $\lambda_{a}$ is utilized for weighting. The notation LN refers to layer normalization.

### 3.2. Attribute Guide Module

In the person Re-ID task, the detailed feature representation of an individual is vulnerable to visual biases introduced by various factors, including camera configurations, viewpoints, lighting conditions, and other environmental variables. These biases can impede the model’s ability to accurately identify the object individually, with such challenges being particularly pronounced in nighttime scenarios. Empirical evidence indicates that integrating attribute information, such as camera IDs and person attributes, can enhance the robustness of feature representations and improve the model’s discriminative capabilities. The Side Information Embedding (SIE) module has been integrated into the existing TransReID framework to incorporate camera ID and viewpoint information into feature representations via a fixed encoding approach, thereby enhancing the discriminant and robustness of these features. Nevertheless, the fixed nature of this encoding method poses a limitation, as it remains invariant to changes in input, thereby hindering the effective acquisition of contextual information.

Inspired by this, we propose the Adaptive Gradient Mechanism (AGM) to dynamically adjust embedding vectors, thereby enhancing adaptability to diverse data distributions and feature spaces. This approach leverages attribute information to generate distinct embedding vectors for each input patch, facilitating improved model comprehension and processing of pedestrian images across varying cameras, viewpoints, and nighttime conditions. It is important to highlight that the AGM is capable of encoding camera and viewpoint information while also allowing for the flexible integration of additional attribute data, such as weather conditions and timestamps, to further enhance the feature representation. The specific AGM is computed as follows:(6)$$\begin{array}{cc} a\left(C,T_{c}\right) & =\sin\left(C/10000^{2T_{c}/D}\right) \end{array}$$
where $C\in Z^{+}$ denotes the number of the currently computed category of auxiliary information, and $T\in Z^{+}$ represents the specific information corresponding to the C of the currently input patch.

Considering that SIE also offers valuable information, the combined utilization of SIE with our AGM can significantly enhance the model’s feature representation capabilities. This approach mitigates the risk of information loss or distortion that might arise from relying on a single method. Furthermore, it preserves AGM’s ability to dynamically adjust the embedding vector, thereby providing the flexibility needed to adapt to varying data distributions and feature spaces. The Embedding Module dynamically adjusts the embedding vectors, thereby enabling flexible adaptation to varying data distributions and feature spaces. Concurrently, the SIE Module offers stable and effective boundary information fusion through fixed encoding. The integration of these two modules facilitates enhanced semantic information fusion. Experiments show that combining the AGM and SIE results in improved recognition accuracy, more stable model performance under different camera viewpoints, lighting conditions, and nighttime scenarios, as well as improved differentiation and consistency of feature representations. In conclusion, the integration of the AGM into the existing SIE module enables comprehensive utilization of camera perspectives and additional attribute data, thereby substantially enhancing the performance of the Transformer model in the person Re-ID task.

### 3.3. Feature Discovery Transformer Framework

The Transformer-based Re-ID methods emphasize the extraction of detailed information. However, the high-frequency components present in nighttime pedestrian images are insufficiently abundant, leading to their potential loss during the optimization process due to the predominance of more abundant low-frequency components.

To address this problem, the Feature Discovery Transformer (FDT) framework uses a dual-branch network to tackle ViT’s tendency to miss details while extracting global features. The main branch directly extracts features from nighttime pedestrian images, enhanced by AGM. Meanwhile, the auxiliary branch uses FRM with Discrete Wavelet Transform (DWT) on stacked images to extract both low- and high-frequency components. The low-frequency components capture the image’s main structure, while high-frequency components hold the details. Features are extracted and flattened by counting paradigms in the high-frequency data. A quantization table is then created for quantization operations. The image components (Y, Cb, Cr) are quantized and inverse-quantized to adjust the image data. The quantized Y, Cb, and Cr components are merged into a complete RGB image, with adjustments for dimensions and normalization to ensure the data are in the correct range. The Inverse Wavelet Transform combines the processed RGB image with high-frequency details, preserving the structure and reducing redundancy, to reconstruct an image that enhances feature extraction and model performance.(7)$$\begin{array}{cc} D\left(Q(L),\alpha \right) & =\left\lfloor \frac{Q(L)+0.5}{\alpha }\right\rfloor \times \alpha \end{array}$$
where Q(L) represents the low-frequency component of the pedestrian images. α is the interval length for finding the nearest quantization point of Q(L); if α is too large, low-frequency information may be lost. Then the Q(L) is fused with the high-frequency portion of the pedestrian image that has not been additionally processed to obtain a new reconstructed pedestrian image. The reconstructed pedestrian image and the original pedestrian image are used as inputs for the auxiliary branch and the main branch, respectively. To avoid over-smoothing by high-frequency components, we apply Normalized Contrastive Loss (NCL) in the auxiliary branch to replicate the *y* from the main branch, receiving the same input $y^{\prime }$. This results in the feature embedding $Y^{\prime }\in R^{(N+1)\times D}$, while the auxiliary branch’s reconstructed image also produces feature embeddings $Y^{r}\in R^{(N+1)\times D}$. The two sets of feature embeddings are fed into the NCL. Following an initial transformation involving basic dimensional rearrangement, the data along the final dimension undergoes normalization using the L2 norm. Subsequently, a similarity matrix is derived by computing the dot product of the two normalized feature vectors. Therefore, a mask matrix is constructed, wherein specific positions are assigned a value of 1, while all other positions are set to 0. The diagonal part of the similarity matrix is extracted from the mask matrix. The exponential function is applied to the diagonal and total similarity of the similarity matrix with temperature scaling, and finally, the NCL is calculated, the specific NCL calculation process is as follows:(8)$$\begin{array}{cc} S(i,j,n) & =\exp\left(\frac{{Y^{\prime }}_{i,n}}{{∥Y_{i,n}^{\prime }∥}_{2}}\cdot \frac{Y_{j,n}^{r}}{{∥Y_{j,n}^{r}∥}_{2}}\right) \end{array}$$(9)$$L_{NCL}=-\frac{1}{N}\sum_{n\in \Omega }\frac{1}{B}\sum_{i=1}^{B}\left[\log\frac{1}{M}\sum_{j:l_{j}=l_{i}}S\left(i,j,n\right)-\log\left(\frac{1}{M}\sum_{j:l_{j}=l_{i}}S\left(i,j,n\right)+\frac{\tau }{B-M}\sum_{j:l_{j}\neq l_{i}}S\left(i,j,n\right)\right)\right]$$
where N is the number of images per ID in the batch. $Y_{i,n}^{\prime }$ is the i-th sample in $Y^{\prime }$, while $Y_{j,n}^{r}$ denotes the j-th positive sample of $Y^{r}$, and j is that of the negative sample. S(i,j,n) is to compute the cosine similarity of the two vectors. τ is the temperature parameter.

While the loss functions of the FDT model consist of ID loss, Triplet loss, and the weights of both, our FDT framework does not fuse the features extracted from the two branches but uses the loss to constrain the features of the main branch as follows:(10)$$\begin{array}{cc} L_{ID} & =L_{ID}^{AGM}+L_{ID}^{FRM}+L_{ID} \end{array}$$(11)$$\begin{array}{cc} L_{Tri} & =L_{Tri}^{AGM}+L_{Tri}^{FRM}+L_{Tri} \end{array}$$(12)$$\begin{array}{cc} L & =L_{ID}+L_{Tri} \end{array}$$

## 4. Experiments

### 4.1. Datasets and Evaluation Metrics

We conduct experiments on two datasets: **Night600** [24] and **RGBNT201**_*rgb*_ [25]. The Night600 dataset comprises 28,813 images representing 600 distinct identities, captured from eight different camera viewpoints. For the training phase, 14,462 images corresponding to 300 identities were employed, while the testing phase utilized a gallery of 14,351 images from the remaining 300 identities. From each viewpoint of each identity, three images were randomly selected to serve as a query set, with the remaining images constituting the gallery set. Consequently, a total of 2180 probe images were obtained to form the query set. The RGBNT201 dataset comprises 3951 images representing 171 distinct identities, captured across three well-aligned modalities. In our evaluation of model performance, we exclusively utilize data from the RGB modality, i.e., ${\mathrm{RGBNT}201}_{rgb}$. This subset includes 836 images from 30 identities designated for training, and an additional 836 images from the same 30 identities are used in the gallery for testing purposes.

To evaluate our model, we employ the Cumulative Matching Characteristic (CMC) curve, and mean Average Precision (mAP) is used to evaluate the performance of our model. The CMC metric is reported via Rank-1, Rank-5, and Rank-10.

### 4.2. Implementation Details

The implementation of our proposed method is conducted using the PyTorch framework (ver 2.5.1). For the baseline, we utilize a ViT-B/16 model that has been pre-trained on the extensive unlabeled dataset LUPerson [57]. The feature dimensionality is set to D=384. The input images are resized to 384×128, with data augmentation techniques such as random horizontal flipping, cropping, and erasing applied. A patch size of P=16 is utilized, whereby each feature map generated following the IBN-based convolutional stemming process is divided into patches measuring 8×8. For all our experiments, we utilized a GEFORCE RTX 3090 GPU. The training process was conducted for a maximum of 180 epochs. The initial learning rate was established at 0.0004, with a warm-up period spanning 20 epochs. Each batch comprised 16 input images, and a weight decay parameter of $1\times 10^{-4}$ was applied. The cosine annealing strategy was employed for learning rate scheduling. The AGM integration information is assigned a weight of 0.55. The auxiliary branch employs a uniform weight parameter of 0.1 for its loss calculations. Additionally, the Triplet Loss and ID Loss within the auxiliary branch are adjusted by multiplying the current iteration number by the minimum of 0.05 and 0.5. The quantization parameter for the low-frequency component is set to α=5. Furthermore, the temperature parameter τ in the NCL framework is set to 4.

### 4.3. Comparisons with State-of-the-Art Methods

We compared three types of advanced methods, CNN-based, Transformer-based and CNN-Transformer-based methods, on two realistic nighttime person datasets, i.e., Night600 [24] and ${\mathrm{RGBNT}201}_{rgb}$ [25]. The CNN-based methods include IDE* [58], PCB [59], MGN [60], ABDNet [61], BoT [36], AGW [37], CCSFG [62], IICI [63], FastReID [64], and IDF [24]. The Transformer-based methods contain TransReid [32], PASS [34], DC-Former [65], IICI (Vit) [63], and PSD [66]. The CNN-Transformer methods consist of HAT [67] and NFormer [68]. To ensure fair experimental results, each method was trained and tested on both datasets without using a re-ranking operation.

As shown in Table 1, ABDNet [61] and TransReid [32] achieved state-of-the-art results on Night600 [24]. However, as illustrated in Table 2 on ${\mathrm{RGBNT}201}_{rgb}$ [25], only TransReid [32] maintained optimal performance. We conducted experiments on two CNN-Transformer-based methods. From the experimental results, it is evident that these two methods were originally designed for daytime scenarios and involved relatively minor modifications to the CNN architecture. Consequently, when compared with CNN-based methods, the performance difference among them is relatively small. Moreover, it can be observed that the Transformer-based method demonstrates superior performance for the nighttime datasets. Therefore, our backbone is to select the Transformer-based network for nighttime person Re-ID research. Therefore, TransReid was employed as the baseline for our study. The enhanced experimental outcomes demonstrate an improvement of 0.4% mAP on the Night600 dataset and a 1.1% increase on the ${\mathrm{RGBNT}201}_{rgb}$ dataset [25] in comparison to TransReid.

### 4.4. Ablation Study

To verify the effectiveness of each component in our proposed FDT framework, we conduct ablation studies, including AGM, FRM, and NCL. The AGM ablation study is shown in Figure 4. And all ablation experiments are on the Night600 [24] and ${\mathrm{RGBNT}201}_{rgb}$ [25] datasets, and the results are shown in Table 3.

**Effectiveness of AGM.** Compared to the baseline method, the combination of FRM and AGM shows a performance improvement of 0.2% on ${\mathrm{RGBNT}201}_{rgb}$. However, this improvement remains inferior to the performance achieved by FDT and is suboptimal when evaluated on the Night600 dataset. Figure 4 shows that AGM is also effective on the Night600 dataset.The Night600 dataset is challenging due to blurry images caused by low-light conditions, making it hard to distinguish pedestrians from the background. Without auxiliary information, the network struggles with noise, making such information essential for training. The proposed AGM effectively addresses this issue by dynamically incorporating limited labeling information to enhance the network’s ability to mitigate the impact of extraneous information, thereby achieving feature enhancement. As illustrated in Figure 4, the AGM demonstrates improved performance across Rank1, Rank5, Rank10, and mAP. In comparison to the baseline model, the attribute information integrated by the AGM significantly enhances the extracted features, reduces the influence of irrelevant information on pedestrian features, and stabilizes the model’s performance. AGM serves to mitigate the impact of appearance-irrelevant features. It does so by dynamically embedding attribute information sourced from the dataset into the model. Nevertheless, neither the Night600 dataset nor the ${\mathrm{RGBNT}201}_{rgb}$ dataset furnishes adequate attribute details. These details, which include aspects like pedestrians’ clothing, weather conditions, and hairstyles, are lacking. As a result, the performance enhancement achieved by AGM remains constrained.

**Effectiveness of AGM, FRM, and NCL.** It can be observed that while AGM is effective in diminishing the presence of irrelevant features, its direct application with FRM fails to address the issue of over-smoothing high-frequency components, and furthermore, FRM introduces interference. To address this challenge, the novel NCL is introduced. Since NCL computation relies on FRM reconstruction, the incorporation of FRM is essential for the implementation of NCL. Meanwhile, it can be seen that, in comparison to AGM and FRM without NCL, the final performance exhibits an enhancement of 0.9% mAP, with an improvement of 0.2% Rank1 and 0.9% Rank10. Therefore, the integration of FRM and NCL proves effective in managing both low- and high-frequency components, thereby preventing the overfitting of low-frequency components and the over-smoothing of high-frequency components by low-frequency components. As shown in Table 3, when only the FRM and NCL modules are employed, the performance actually declines. The reason is that while FRM and NCL enhance pedestrian features and reduce the loss of high-frequency components during the processing of low-frequency components, they also cause the auxiliary branch to be unable to handle the noise in high-frequency components. Additionally, the main branch, where AGM is added, weakens the influence of appearance-irrelevant features. Therefore, the pedestrian features extracted with AGM, FRM, and NCL could improve the performance of the nighttime person Re-ID task.

### 4.5. Visualizations

**Visualization of Attention Map**. Figure 5 illustrates the attention map of FDT and the baseline on Night600. The results indicate that, in nighttime scenarios, the baseline model struggles to accurately ascertain the positions of pedestrians. Consequently, the model’s attention is inadequately focused on pedestrians, and it fails to effectively mitigate the influence of background noise and other extraneous information. This limitation results in the extraction of pedestrian identity features being compromised by irrelevant features. Our FDT can identify a pedestrian’s body position and attention, providing a clearer attention heat map than the naked eye. However, it still misses details like clothing, focusing solely on the human body. AGM fails to obtain attribute information regarding clothing details from the dataset. Consequently, it cannot strengthen the learning of pedestrian clothing features. On the other hand, FRM and NCL primarily focus on the learning of low-frequency components. This enables the model to adequately learn the main structure of the image. Simultaneously, through the processing of low-frequency components, the model reduces the over-smoothing of high-frequency components. However, it does not explicitly enhance the high-frequency components. As a result, for some data with clothing details, FDT is unable to perform recognition, and attention to clothing is decreased. Figure 6 shows that on ${\mathrm{RGBNT}201}_{rgb}$, some data are better lit, yet the baseline is distracted by irrelevant features, focusing on the foreground. Our FDT is more robust against appearance-irrelevant features. It can better focus on the person’s body. Although there is still room for improvement in the attention to detailed features such as clothing, compared to other methods, our FDT shows stronger attention to these details. Therefore, FDT concentrates on pedestrians, capturing detailed high-frequency features more effectively.

**Top-10 retrieval results.** Figure 7 and Figure 8 present the top-10 retrieval results between the FDT model and the baseline on Night600 and ${\mathrm{RGBNT}201}_{rgb}$, respectively. It is evident that the FDT model continues to experience some degree of environmental interference. However, compared to the baseline, the accuracy is improved. Although the FDT model is less susceptible to irrelevant information caused by environmental factors, such as street lights and incorrect pedestrian images, it remains inevitably affected to some extent in the real world.

**t-SNE.** Figure 9 and Figure 10 present the t-SNE visualization, effectively illustrating the intra- and inter-class distances among the 20 classes within Night600 and ${\mathrm{RGBNT}201}_{rgb}$, respectively. The t-SNE visualization demonstrates that FDT effectively reduces intra-class distances while increasing inter-class distances. This suggests that FDT proficiently differentiates between features of distinct individuals while concurrently consolidating features pertaining to the same individual. When compared to the t-SNE distribution of the Baseline method, our FDT uses the NCL to increase the distance between classes. However, due to the fact that the noise in the high-frequency components and the pedestrian identity-relevant features are not processed separately, some classes are incorrectly drawn closer together. This issue is particularly prominent in the much more challenging Night600 dataset. This is the reason why there is still substantial room for improvement in the overall performance of our FDT.

**Visualization of Loss and Accuracy.** Figure 11 and Figure 12 show the variations in loss and accuracy of our FDT model during training on the Night600 and ${\mathrm{RGBNT}201}_{rgb}$ datasets, respectively. The results indicate that the curves on Night600 exhibit a smoother trajectory. In contrast, the ${\mathrm{RGBNT}201}_{rgb}$ dataset, characterized by significant variations in data illumination, displays initially steep curves that eventually stabilize. This stabilization demonstrates that our method achieves state-of-the-art performance.

## 5. Conclusions

In this paper, we propose a new Feature Discovery Transformer (FDT) model using the “enhancement drop” manner for nighttime person Re-ID. The Frequency-wise Reconstruction Module (FRM) quantizes the low-frequency components of the image separately to cope with the overfitting during the learning process of the low-frequency components of the image, then combines it with the original high-frequency components to reconstruct it and preserves the detailed information contained in the high-frequency components through Normalized Contrastive Loss (NCL) to learn more pedestrian identity features. Meanwhile, the Attribute Guide Module (AGM) is proposed to integrate the auxiliary information and reduce the influence of appearance-irrelevant features to make the learned pedestrian identity features more discriminative. Experiments show that our FDT on Night600 and ${\mathrm{RGBNT}201}_{rgb}$ datasets has outperformed the state-of-the-art performance of existing person Re-ID methods.

Nevertheless, there are still some limitations in the FDT: (1) AGM relies on the amount of auxiliary information provided by the dataset; however, the auxiliary information of existing datasets is very limited. (2) When FRM separates the low-frequency and high-frequency components, there is still room for processing the high-frequency components. These issues will be investigated in the future.

## Figures and Tables

**Figure 1 sensors-25-00862-f001:**
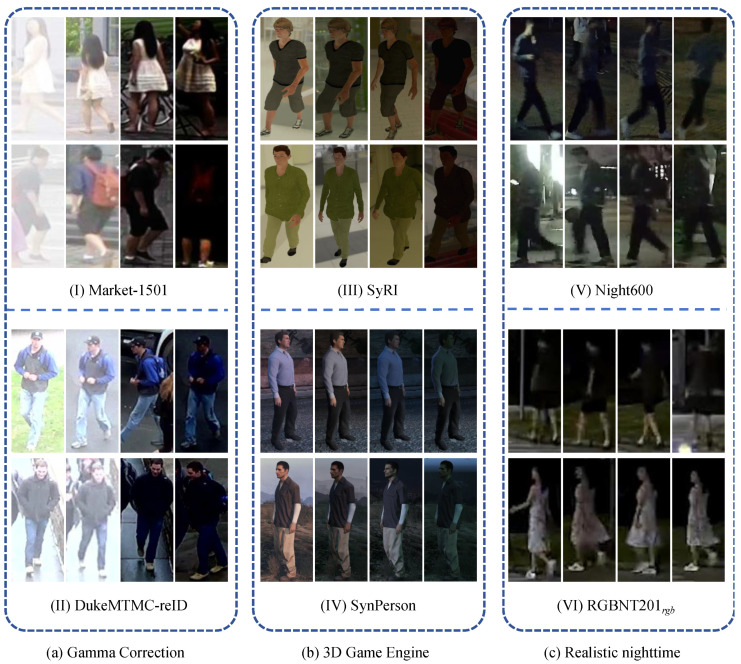
Illustration of image samples from low-illumination person Re-ID datasets. (**a**) Synthesized low-illumination datasets using gamma correction, containing (**I**) Market-1501 [19] and (**II**) DukeMTMC-reID [20]. (**b**) Synthesized low-illumination datasets using 3D game engines: (**III**) SyRI [21] and (**IV**) SynPerson [22]. (**c**) Realistic nighttime datasets: (**V**) Night600 [24] and (**VI**) ${\mathrm{RGBNT}201}_{rgb}$ [25].

**Figure 2 sensors-25-00862-f002:**
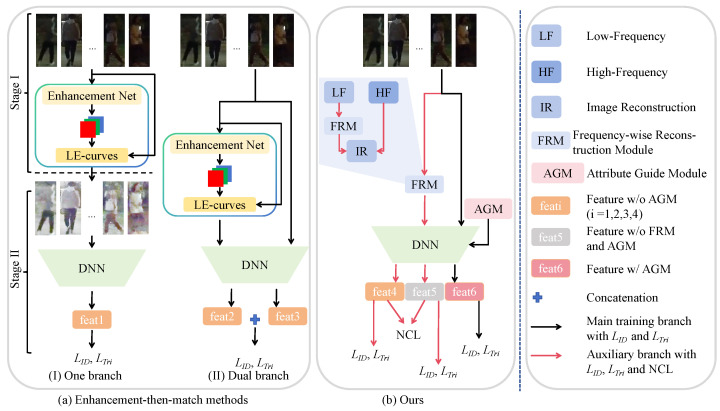
Comparison with current routine nighttime person Re-ID methods. (**a**) are “enhancement-then-match” methods, which have two manners: (**I**) one branch and (**II**) dual branch. In particular, the one branch consists of two stages, corresponding to enhancement and Re-ID processes. The dual branch is an end-to-end nighttime person Re-ID module, including enhancement and Re-ID processes. (**b**) is our proposed FDT model, which is simpler and more effective than other methods in an “enhancement drop” manner.

**Figure 3 sensors-25-00862-f003:**
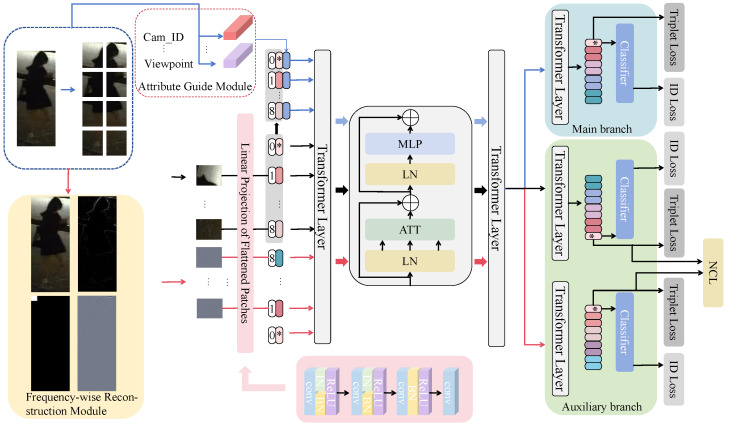
The overall of our proposed FDT framework, which consists of FRM and AGM. Note that the FDT model uses ViT as the backbone, and the extraction of nighttime pedestrian image features serves as the main branch, while the recombination of extracted high- and low-frequency components captured by FRM into pedestrian image features functions as the auxiliary branch. Additionally, the features enhanced with high- and low-frequency components are employed to constrain the main branch, alongside the nighttime pedestrian features that are not extracted using the AGM.

**Figure 4 sensors-25-00862-f004:**
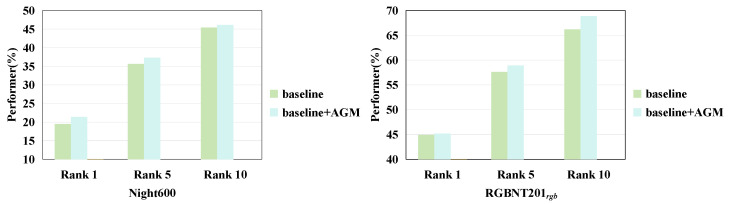
Evaluate the performance comparison of the baseline and FDT methods on the Night600 [24] and ${\mathrm{RGBNT}201}_{rgb}$ [25] after ablating the AGM. Rank1 (%), Rank5 (%), Rank10 (%) are reported.

**Figure 5 sensors-25-00862-f005:**
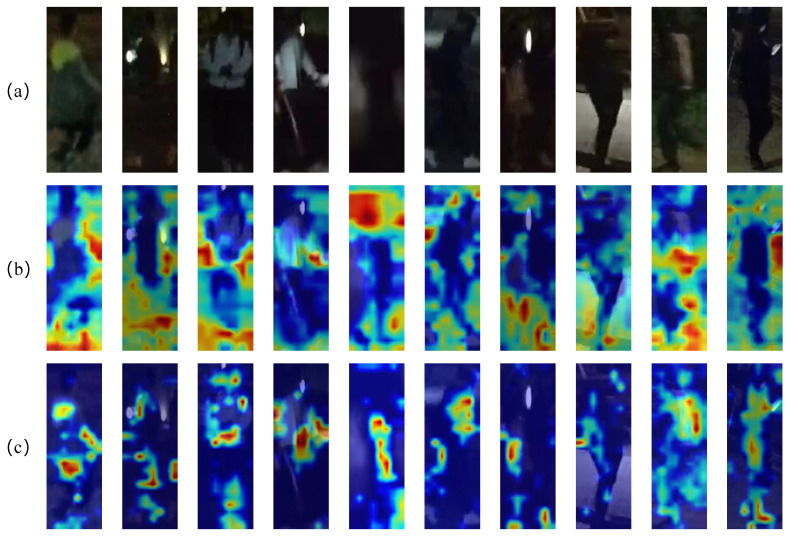
(**a**) Person image from Night600 data [24] shown using Grad-CAM to get the attention maps for the visualization of (**b**) Baseline and (**c**) Our proposed FDT.

**Figure 6 sensors-25-00862-f006:**
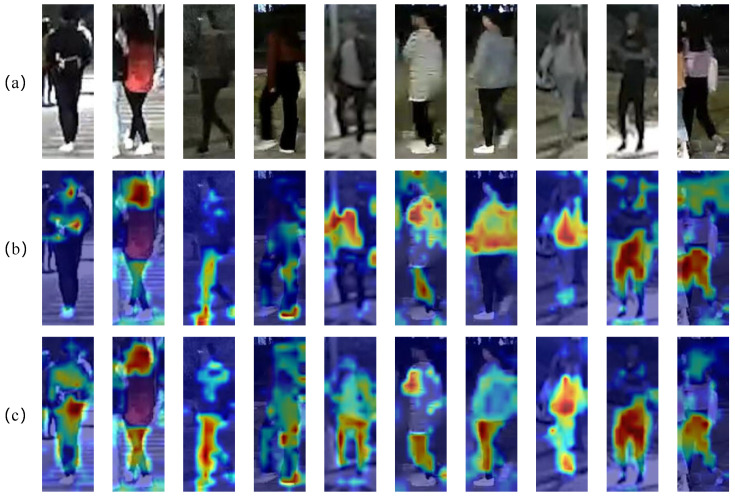
(**a**) Person image from ${\mathrm{RGBNT}201}_{rgb}$ dataset [25] shown using Grad-CAM to get the attention maps for the visualization of (**b**) Baseline and (**c**) Ours proposed FDT.

**Figure 7 sensors-25-00862-f007:**
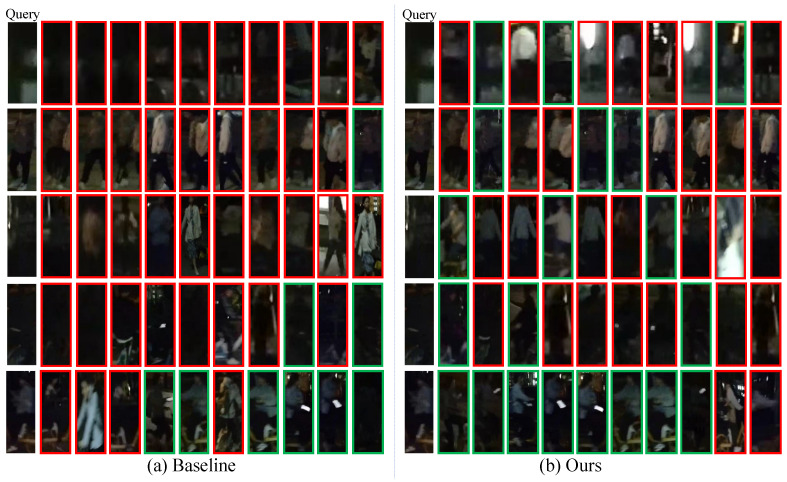
Illustration of top-10 retrieval results obtained by Baseline and Ours FDT method on the Night600. The green box indicates a correct match, whereas the red box is the opposite.

**Figure 8 sensors-25-00862-f008:**
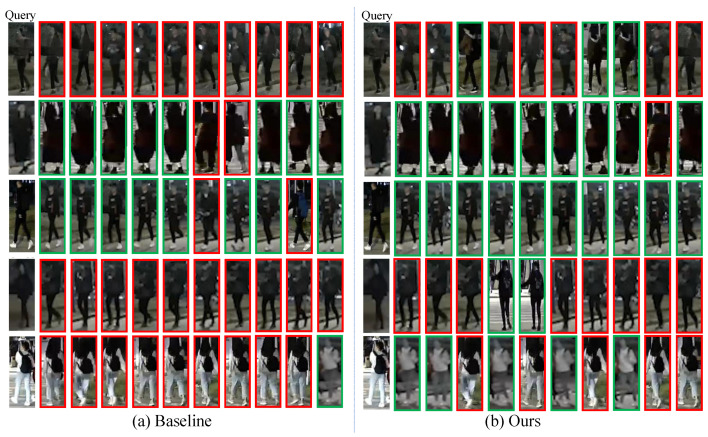
Illustration of top-10 retrieval results obtained by Baseline and Ours FDT method on the ${\mathrm{RGBNT}201}_{rgb}$. The green box indicates a correct match, whereas the red box is the opposite.

**Figure 9 sensors-25-00862-f009:**
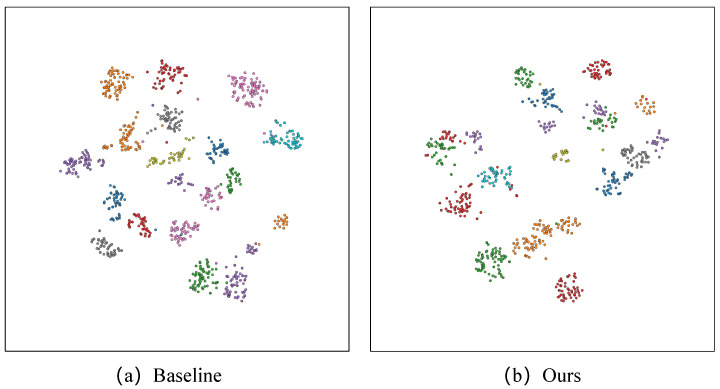
The t-SNE visualization between baseline and our FDT model on the Night600.

**Figure 10 sensors-25-00862-f010:**
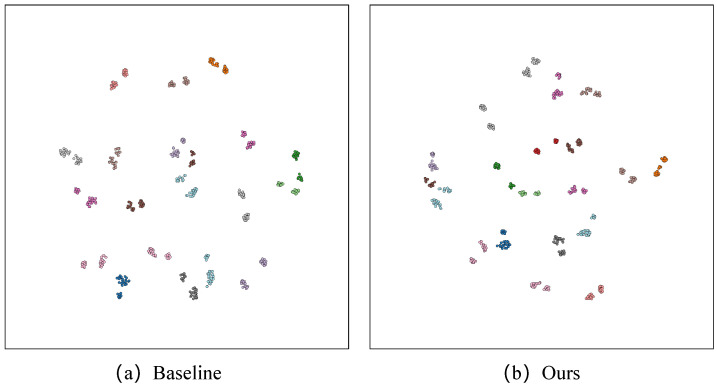
The t-SNE visualization between baseline and our FDT model on the ${\mathrm{RGBNT}201}_{rgb}$.

**Figure 11 sensors-25-00862-f011:**
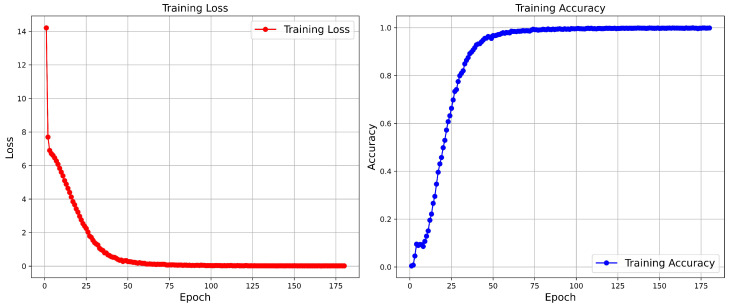
Loss and accuracy curve of our proposed FDT model on the Night600.

**Figure 12 sensors-25-00862-f012:**
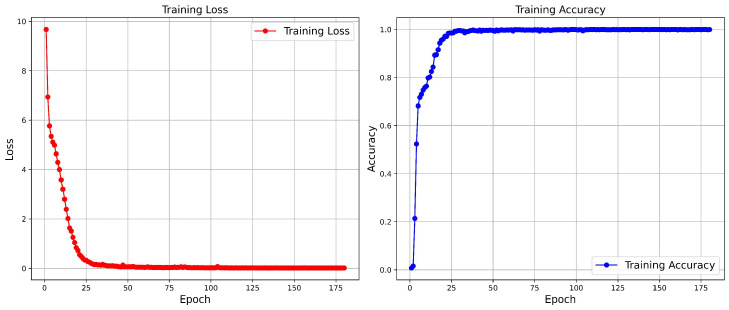
Loss and accuracy curve of our proposed FDT model on the ${\mathrm{RGBNT}201}_{rgb}$.

**Table 1 sensors-25-00862-t001:** Comparison with existing state-of-the-art methods on Night600 datasets. Rank1 (%), Rank5 (%), Rank10 (%) and mAP (%) are reported. The best and second results are highlighted in **bold** and underline, respectively.

			Night600			
Backbone	Method	Venue	Rank1	Rank5	Rank10	mAP
CNN	IDE* [58]	TOMM17	10.4	21.6	29.0	3.3
	PCB [59]	ECCV18	11.4	22.7	30.6	5.9
	MGN [60]	ACM MM18	5.4	10.3	15.2	3.9
	ABDNet [61]	ICCV19	15.5	32.2	42.2	7.9
	BoT [36]	CVPRW19	12.4	23.9	30.7	5.2
	AGW [37]	TPAMI21	13.6	23.9	32.4	6.4
	CCSFG [62]	CVPR22	12.2	26.2	34.2	5.4
	IICI [63]	ACM MM23	8.4	19.2	25.6	3.1
	FastReID [64]	ACM MM23	14.6	26.6	33.9	7.2
	IDF [24]	TMM23	11.8	27.7	36.6	6.1
Transformer	TransReid [32]	ICCV21	14.9	28.0	38.5	7.5
	PASS [34]	ECCV22	12.8	28.7	37.6	6.7
	DC-Former [65]	AAAI23	8.9	19.8	29.1	5.9
	IICI(ViT) [63]	ACM MM23	11.2	22.9	29.5	4.3
	LRIMV [69]	TNNLS23	5.4	12.2	17.3	2.9
	PSD [66]	ICCV23	7.3	17.2	22.8	3.4
	TransReid-SSL [54]	arxiv21	19.5	35.6	**45.4**	9.1
CNN-Transformer	HAT [67]	MM21	6.4	13.0	19.2	3.1
	NFormer [68]	CVPR22	10.0	22.9	31.8	4.3
Transformer	FDT (Ours)	-	**19.9**	**36.0**	**45.4**	**9.5**

**Table 2 sensors-25-00862-t002:** Comparison with existing state-of-the-art methods on ${\mathrm{RGBNT}201}_{rgb}$ datasets. Rank1 (%), Rank5 (%), Rank10 (%) and mAP (%) are reported. The best and second results are highlighted in **bold** and underline, respectively.

			${\mathrm{RGBNT}201}_{\mathrm{rgb}}$			
Backbone	Method	Venue	Rank1	Rank5	Rank10	mAP
CNN	IDE* [58]	TOMM17	10.4	21.6	29.0	3.3
	PCB [59]	ECCV18	10.0	19.3	26.6	15.4
	MGN [60]	ACM MM18	16.2	26.0	33.4	21.5
	ABDNet [61]	ICCV19	18.9	32.5	44.55	22.5
	BoT [36]	CVPRW19	17.5	30.1	39.9	21.0
	AGW [37]	TPAMI21	21.1	37.2	49.3	23.7
	CCSFG [62]	CVPR22	27.7	38.8	48.9	28.9
	IICI [63]	ACM MM23	1.6	3.4	5.1	4.4
	FastReID [64]	ACM MM23	24.9	39.0	48.6	27.2
Transformer	TransReid [32]	ICCV21	37.2	54.1	64.5	38.2
	PASS [34]	ECCV22	20.3	34.9	46.0	24.9
	DC-Former [65]	AAAI23	9.8	22.4	34.8	15.1
	IICI(Vit) [63]	ACM MM23	34.6	50.1	59.4	34.7
	LRIMV [69]	TNNLS23	38.2	55.7	64.7	38.1
	PSD [66]	ICCV23	28.8	43.0	52.9	30.6
	TransReid-SSL [54]	arxiv21	44.9	57.6	66.2	45.8
CNN-Transformer	HAT [67]	MM21	13.1	26.3	35.7	19.8
	NFormer [68]	CVPR22	19.0	31.8	41.6	20.3
Transformer	FDT (Ours)	-	**45.0**	**59.6**	**70.4**	**46.9**

**Table 3 sensors-25-00862-t003:** Ablation studies on Night600 and ${\mathrm{RGBNT}201}_{rgb}$ datasets. Rank1 (%), Rank5 (%), Rank10 (%) and mAP (%) are reported. The best and second results are highlighted in **bold** and underline, respectively.

AGM	FRM	NCL	Night600				${\mathrm{RGBNT}201}_{\mathrm{rgb}}$			
			Rank1	Rank5	Rank10	mAP	Rank1	Rank5	Rank10	mAP
-	-	-	19.5	35.6	**45.4**	9.1	44.9	57.6	66.2	45.8
-	✓	✓	17.6	33.5	42.8	8.7	37.3	51.4	63.8	38.7
✓	✓	-	19.7	35.6	44.5	8.4	43.7	58.0	69.3	46.0
✓	✓	✓	**19.9**	**36.0**	**45.4**	**9.5**	**45.0**	**59.6**	**70.4**	**46.9**

## Data Availability

For the Night600 dataset, register a GitHub account and log in, then find the download link on this page: https://github.com/Alexadlu/IDF (accessed on 10 April 2023). The ${\mathrm{RGBNT}201}_{rgb}$ dataset is also available for download at the above link (accessed on 18 May 2021).

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
