# Peer review of "Identity Hides in Darkness: Learning Feature Discovery Transformer for Nighttime Person Re-Identification"

_sensors, 2025, doi:10.3390/s25030862_

Round 1

Reviewer 1 Report

Comments and Suggestions for Authors

The article presents the Feature Discovery Transformer (FDT) method to enhance pedestrian identification at night by employing frequency decomposition and adaptive feature learning to improve accuracy under low-light conditions. The topic of the article is relevant. However, the article's structure does not align with the MDPI format for research articles (Introduction, including a review of related works, Models and Methods, Results, Discussion, Conclusions). The level of English is acceptable, and the text is easy to read. The figures are of reasonable quality. The article cites 70 sources, some of which are outdated, and the References section is formatted carelessly.

The following remarks and recommendations can be made regarding the article's content:

1.The study introduces a new method, FDT, which integrates the Frequency-wise Reconstruction Module (FRM), the Attribute Guide Module (AGM), and Normalized Contrastive Loss (NCL). However, the authors do not sufficiently describe how these components interact to achieve the stated results. For example, the relationship between frequency decomposition and the improvement of pedestrian identification features is underexplored. For MDPI articles of the Essay type, it is essential for authors to provide detailed explanations of how the proposed method addresses the identified challenges, such as poor image quality or noise, at a theoretical level and then demonstrate this in practice.

2.The "Related Works" section provides an overview of existing methods, such as "enhancement-then-match" approaches and transformer architectures for Re-ID. However, it lacks a critical analysis demonstrating the limitations of prior approaches. For instance, the weaknesses of using synthetic data in previous works or the constraints of modern transformers under nighttime conditions are not discussed.

3. The conclusion briefly mentions the limitations of the method, such as the dependency of AGM on additional attributes and the need for improved handling of high-frequency components. These aspects should be elaborated upon in the main text, linking them to the experimental results. For example, the model's insufficient accuracy under certain conditions (e.g., RGBNT201rgb dataset) could be explained by dataset or model architecture limitations.

4. The authors focus on only two datasets (Night600 and RGBNT201rgb), limiting the understanding of the model's applicability in other scenarios, such as surveillance or multispectral tasks. Additionally, they do not discuss how FDT might adapt to other types of data, such as images with varying viewing angles or different noise densities.

5. While the authors note that existing datasets lack paired data for training or have limited meta-informational attributes, they do not consider how these limitations affect the experimental results. For example, to what extent does the absence of certain data reduce the model's performance?

6. The article includes visualizations such as attention maps and top-10 retrieval results, but their interpretation is superficial. For instance, the attention maps show that the model focuses on the human body while ignoring clothing details, but this is not discussed. Moreover, the lack of comparisons with competing methods limits the understanding of FDT's advantages.

7. All experiments are conducted on two datasets, which, while relevant to the topic, do not demonstrate the method's flexibility and adaptability in various scenarios, such as dynamic conditions (video data) or different types of cameras.

Reviewer 2 Report

Comments and Suggestions for Authors

Review Comments: The paper introduces a novel Feature Discovery Transformer framework designed for nighttime person Re-ID. It integrates two key modules: the Frequency-wise Reconstruction Module, which separates and processes high- and low-frequency image components, and the Attribute Guide Module, which incorporates auxiliary information to enhance feature robustness. The model avoids traditional image enhancement methods, focusing instead on directly extracting discriminative identity features from nighttime images. Experiments on two benchmarks, Night600 and RGBNT201rgb, demonstrate that FDT achieves state-of-the-art performance in nighttime person Re-ID tasks. In general, the motivation for this submission is easy to understand and the insight is interesting. However, there are still several weaknesses, as follows:

  1. The dual-branch approach introduces significant computational overhead compared to single-branch models, but the article does not quantify this trade-off. Parameters such as the quantization interval (α) and temperature (τ) are predefined but lack sensitivity analysis.
  2. Transformer-based methods (e.g., TransReID-SSL) are used as baselines, but the paper does not compare against ensemble methods or those combining CNN and transformers.
  3. Some visualizations, such as Grad-CAM and t-SNE, are included without sufficient accompanying analysis, making it hard to interpret their implications fully. In addition, the ablation study lacks nuanced interpretations. For instance, while FRM and AGM show some improvements individually, their combined impact is not thoroughly dissected.
  4. Certain cited works lack sufficient detail, such as publication years or full journal names. In addition, some works about image recognition and Re-ID are suggested to be cited in this paper to make it more comprehensive, such as 10.1109/TCSVT.2023.3339167

Reviewer 3 Report

Comments and Suggestions for Authors

There are a lot of sentences that are difficult to understand.

For example:

Line 316: “Features are extracted and flattened by counting paradigms in the high-frequency data.” What is “counting paradigms”? How does it work?

Line 330: “To avoid over-smoothing by high-frequency components.” What does that mean?

FRM is the contribution point mentioned in the paper, but from Table 3, it can be seen that when only the FRM and NCL modules are used, the performance actually decreases? It is necessary to leverage prior information from AGM to improve the metrics.

The paper mentions using the AGM module to optimize the previous SIE module, but there is no noticeable improvement in recognition performance when using AGM compared to using only the SIE module.

Round 2

Reviewer 1 Report

Comments and Suggestions for Authors

I have formulated the following comments on the previous version of the article:

1.The study introduces a new method, FDT, which integrates the Frequency-wise Reconstruction Module (FRM), the Attribute Guide Module (AGM), and Normalized Contrastive Loss (NCL). However, the authors do not sufficiently describe how these components interact to achieve the stated results. For example, the relationship between frequency decomposition and the improvement of pedestrian identification features is underexplored. For MDPI articles of the Essay type, it is essential for authors to provide detailed explanations of how the proposed method addresses the identified challenges, such as poor image quality or noise, at a theoretical level and then demonstrate this in practice.

2.The "Related Works" section provides an overview of existing methods, such as "enhancement-then-match" approaches and transformer architectures for Re-ID. However, it lacks a critical analysis demonstrating the limitations of prior approaches. For instance, the weaknesses of using synthetic data in previous works or the constraints of modern transformers under nighttime conditions are not discussed.

3. The conclusion briefly mentions the limitations of the method, such as the dependency of AGM on additional attributes and the need for improved handling of high-frequency components. These aspects should be elaborated upon in the main text, linking them to the experimental results. For example, the model's insufficient accuracy under certain conditions (e.g., RGBNT201rgb dataset) could be explained by dataset or model architecture limitations.

4. The authors focus on only two datasets (Night600 and RGBNT201rgb), limiting the understanding of the model's applicability in other scenarios, such as surveillance or multispectral tasks. Additionally, they do not discuss how FDT might adapt to other types of data, such as images with varying viewing angles or different noise densities.

5. While the authors note that existing datasets lack paired data for training or have limited meta-informational attributes, they do not consider how these limitations affect the experimental results. For example, to what extent does the absence of certain data reduce the model's performance?

6. The article includes visualizations such as attention maps and top-10 retrieval results, but their interpretation is superficial. For instance, the attention maps show that the model focuses on the human body while ignoring clothing details, but this is not discussed. Moreover, the lack of comparisons with competing methods limits the understanding of FDT's advantages.

7. All experiments are conducted on two datasets, which, while relevant to the topic, do not demonstrate the method's flexibility and adaptability in various scenarios, such as dynamic conditions (video data) or different types of cameras.

The authors have addressed all my comments. I found their responses quite convincing. I support the publication of the current version of the article. I wish the authors creative success.

Reviewer 3 Report

Comments and Suggestions for Authors

The author has basically answered my questions, and the paper can be published.